# Faster Perfect Sampling of Bayesian Network Structures

**Juha Harviainen**[1]                    **Mikko Koivisto**[1]

[1]Department of Computer Science, University of Helsinki, Helsinki, Finland

## Abstract

Bayesian inference of a Bayesian network structure amounts to averaging over directed acyclic graphs (DAGs) on a given set of $n$ variables, each DAG weighted by its posterior probability. In practice, save some special inference tasks, one averages over a sample of DAGs generated perfectly or approximately from the posterior. For the hard problem of perfect sampling, we give an algorithm that runs in $O(2.829^n)$ expected time, getting below $O(3^n)$ for the first time. Our algorithm reduces the problem into two smaller sampling problems whose outputs are combined; followed by a simple rejection step, perfect samples are obtained. Subsequent samples can be generated considerably faster. Empirically, we observe speedups of several orders of magnitude over the state of the art.

## 1 INTRODUCTION

Bayesian networks are probabilistic graphical models whose structure, a directed acyclic graph (DAG), encodes conditional independences among the modelled variables. To learn the structure, the score-based approach assigns a score to each possible DAG, roughly quantifying how well it fits the data and background knowledge. Commonly used *modular* scoring functions factorize into a product of node-wise *local scores*, each of which only depends on the node and its parents. This structural property enables finding a globally optimal DAG significantly faster than by exhaustive search, e.g., by dynamic programming [Ott et al., 2004, Singh and Moore, 2005, Silander and Myllymäki, 2006], by related A* search [Yuan et al., 2011], or by linear programming [Bartlett and Cussens, 2017]. The optimization problem being NP-hard [Chickering, 1995], various methods have been proposed for finding local optima, including many recent ones based on continuous optimization [Zheng et al., 2018].

However, outputting a single DAG can be problematic, particularly if there are little data, as numerous DAGs may have almost equally high scores. The Bayesian approach to learning Bayesian networks [Madigan and York, 1995, Heckerman et al., 1995] takes this into account by averaging over multiple models. Computationally, the approach presents a major challenge and has led to the development of Markov chain Monte Carlo methods [Madigan and York, 1995, Grzegorczyk and Husmeier, 2008, Kuipers and Moffa, 2017], which generate a sample of DAGs approximately from the posterior distribution. While these methods often appear to perform well empirically, they lack good, provable accuracy guarantees.

Several results are known for model averaging with accuracy guarantees. By dynamic programming, one can compute the exact marginal posterior probabilities of edges and related features in time $O(2^n n^2)$, where $n$ is the number of nodes [Koivisto and Sood, 2004, Koivisto, 2006]. Moreover, one can sample DAGs exactly from the posterior, hereafter referred to as *perfect sampling*, with negligible overhead [He et al., 2016]. Unfortunately, these methods rely on a specially structured score, which is nonuniform over Markov equivalent DAGs and often considered undesirable. For the more desirable, modular scores, exact computation of marginals [Tian and He, 2009] and perfect sampling [Talvitie et al., 2019] scale as $3^n n^{O(1)}$. The former problem is #P-hard, i.e., at least as hard as counting the satisfying assignments of a given boolean formula [Harviainen and Koivisto, 2023]. Further, the bound $O(3^n)$ has only been beaten using impractical "fast matrix multiplication" [Koivisto and Röyskö, 2020] that has large constant factors in its time complexity. This raises the following question: *Can we obtain a faster, practical algorithm for perfect sampling by avoiding the computation of exact marginals?*

We answer this question in the affirmative. Our approach employs rejection sampling for a union of sets of DAGs, where each set is associated with a partition of the node set into two halves. First, we precompute the total score of DAGs in each of the sets in time $O(2^{3n/2}n) = O(2.829^n)$ by

utilizing an inclusion–exclusion recurrence of Tian and He [2009] and dynamic programming over partitions of subsets of nodes called *root-layerings* [Kuipers and Moffa, 2015, 2017, Talvitie et al., 2019] and *sink-layerings* [Harviainen and Koivisto, 2023]. In the sampling step, we choose one partition of the nodes, and then sample a DAG according to that partition in time $O(2^{n/2}n) = O(1.415^n)$. Multiple partitions may allow sampling the same DAG, so we occasionally need to *reject* the sample and restart the sampling step to ensure that the *accepted* samples come from the posterior distribution. Consequently, the running time of the algorithm is a random variable.

What can we then say about the time complexity? There are DAGs where the probability of *accepting* is roughly $2^{-n}$, and so the sampling step needs to be restarted roughly $2^n$ times on average if all probability mass is on such graphs. This results in the worst-case expected time $O(2.829^n)$ for getting an accepted sample, matching the preprocessing time. Fortunately, the worst case seems to not occur in practice, of which we give both analytical and empirical evidence: We prove that only a constant number of samples are needed until getting an accepted sample on average over all DAGs on $n$ nodes. Empirically, we observe that our algorithm draws samples several orders of magnitude faster than the previous state of the art, with the speedup depending on the sparsity of sampled DAGs.

As an additional contribution, we remark an application of the present work in sampling DAGs under ancestral constraints. The requirement that a given node must be an ancestor of another node is related to causality, and the computation of the normalizing constant of the scores of such DAGs has been studied before [Chen et al., 2015, Pensar et al., 2020]. On the other hand, the existence of a path is a non-modular feature, so mere manipulation of local scores is insufficient for enabling sampling. By applying our optimized version of the sink-layering algorithm of Harviainen and Koivisto [2023] on the forward–backward decompositions of Pensar et al. [2020], we give the first perfect sampling algorithm for the problem, and achieve preprocessing time $O(3^n n)$ and sampling time $O(2^n n)$.

One drawback of our rejection sampling method is the use of subtraction, which may lead to loss of accuracy in the computations. For this reason, the implementations used in the experiments assume that the local scores are integers. Talvitie et al. [2019] discussed the issue of numerical stability and also gave two algorithms for perfect sampling of DAGs that rely only on *monotone* operations—addition and multiplication. Problematically, the preprocessing step of these monotone algorithms uses roughly $4^n$ operations and requires storing at least $3^n$ numbers to the memory. Thus, we argue that non-monotonicity is likely required to achieve reasonable sampling time and memory usage. The time and space complexities of the different algorithms are summarized in Table 1.

## 2 PRELIMINARIES

We start by recalling the basics of Bayesian networks. Then, we will discuss previous work on weighted counting and sampling of DAGs, which our work utilizes.

### 2.1 BAYESIAN NETWORKS

The structure of a Bayesian network is a directed acyclic graph $D = (N, A)$ with a node set $N$ and a set of directed edges $A$ with $n := |N|$. The nodes correspond to the variables of the model whereas the edge structure encodes their conditional independencies—see, for example, the textbook of Koller and Friedman [2009] for a detailed overview of Bayesian networks. We denote the *score* of $D$ by $w(D)$. Commonly used score functions are *modular*, meaning that they decompose into a node-wise product of *local scores*

$$w(D) = \prod_{i \in N} w_i(D_i),$$

where $D_i$ is the set of parents of the node $i$. The score comprises any modular prior distribution of the DAGs, such as the uniform prior or the fair prior [Friedman and Koller, 2003, Eggeling et al., 2019], and possibly the likelihood function, depending on the application. For example, the Bayesian network structure learning problem looks for a DAG $D$ with the maximum score, or equivalently, the maximum posterior probability.

Using just a single model may lead to poor inference results if there is uncertainty about the correct model. The Bayesian approach to structure learning [Madigan and York, 1995, Heckerman et al., 1995] overcomes this by taking the average over multiple structures: For an event of interest $Q$, we have

$$\Pr(Q) = \sum_D w(D) \Pr(Q \mid D) \Big/ \sum_D w(D)$$

with summation over all DAGs. This can be approximated as

$$\frac{1}{K} \sum_{k=1}^{K} \Pr\left(Q \mid D^k\right)$$

for DAGs $D^1, D^2, \ldots, D^K$ sampled with the probability of $D^k = D$ being proportional to $w(D)$. Thus, we seek to solve the following problem:

DAG SAMPLING
    *Input:* A set of nodes $N$ and a modular function $w$.
    *Output:* Sample a DAG $D$ such that $\Pr(D) \propto w(D)$.

### 2.2 ZETA TRANSFORM

In this paper, we often utilize transforms of functions over subset lattices. Let $f$ be a function whose inputs are subsets

Table 1: Summary of the complexities and the properties of the algorithms. Time and space complexities are asymptotic up to polynomial factors in $n$. For our algorithm, we have rounded up the bases of the exponents for easier comparison.

| Reference | Preprocessing | Sampling (best) | Sampling (worst) | Space | Monotone |
|---|---|---|---|---|---|
| Talvitie et al. [2019] | $3^n$ | $2^n$ | $2^n$ | $2^n$ | no |
| Talvitie et al. [2019] | $4^n$ | $2^n$ | $2^n$ | $3^n$ | yes |
| Talvitie et al. [2019] | $4^n$ | $\mathrm{poly}(n)$ | $\mathrm{poly}(n)$ | $4^n$ | yes |
| *this paper* | $2.829^n$ | $1.415^n$ | $2.829^n$ | $2^n$ | no |

of some ground set. Then, its zeta transform $\hat{f}$ is defined by

$$\hat{f}(S) := \sum_{T \subseteq S} f(T),$$

and its inverse is given by the Möbius inversion formula

$$f(S) = \sum_{T \subseteq S} (-1)^{|S \setminus T|} \hat{f}(T).$$

For example, $\hat{w}_i(S)$ is the sum of local scores of the node $i$ over all subsets of $S$.

Given $f(S)$ for all subsets $S$ of a set of $n$ elements, the values of $\hat{f}$ are computable with $O(2^n n)$ additions and multiplications, and vice versa [Yates, 1937, Kennes and Smets, 1990].

## 2.3   COUNTING

We start reviewing previous work by discussing the computation of normalizing constants of subsets of DAGs. They serve as a building block for the sampling algorithms.

Let $h(U)$ with $U \subseteq N$ be the total score of all DAGs on nodes $U$. Tian and He [2009] discovered an inclusion–exclusion formula for computing the values of $h(U)$: As every DAG has at least one sink node, the set of DAGs on $U$ can be seen as an union over sets of DAGs for which $s \in U$ is a sink. This yields the recurrence

$$h(U) = \sum_{\emptyset \neq S \subseteq U} (-1)^{|S|+1} h(U \setminus S) \prod_{i \in S} \hat{w}_i(U \setminus S), \quad (1)$$

which allows computing all values of $h$ in time $O(3^n n)$. Koivisto and Röyskö [2020] have given an asymptotically slightly faster algorithm of time complexity $O(2.985^n)$ that relies on fast matrix multiplication.

## 2.4   SAMPLING

We next recall the details of the sampling algorithm of Talvitie et al. [2019] for weighted DAGs. Their algorithm of solves the DAG SAMPLING problem with preprocessing time $O(3^n n)$ and sampling time $O(2^n n)$.

The algorithm employs *root-layerings* that are partitions of the node set. The root-layering $(R_1, R_2, \ldots, R_\ell)$ of a

DAG $D$ is obtained by letting the first layer $R_1$ contain the source nodes of $D$, the second the sources of the subgraph $D[N \setminus R_1]$ induced by $N \setminus R_1$, and so forth. In general, layer $R_k$ contains nodes $i$ for which the longest path from a node in $R_1$ to $i$ is of length $k - 1$. For notational convenience, let $R_0 = \emptyset$ unless otherwise specified. Root-layerings are illustrated in Figure 1a.

By definition, a node $i \in R_{k+1}$ must have at least one parent in $R_k$, and its other parents are a subset of

$$R_{1:k} := \bigcup_{i=1}^{k} R_i.$$

Denote the total score of this collection of *potential parent sets* by $\hat{w}_i(R_k, R_{1:k})$. Their values can be efficiently queried by using the identity

$$\hat{w}_i(R, S) = \hat{w}_i(S) - \hat{w}_i(S \setminus R)$$

after precomputing the zeta transforms of the local scores in time $O(2^n n^2)$. For convenience, we let $\hat{w}_i(\emptyset, \emptyset) = w_i(\emptyset)$.

Now, the total score of all DAGs with a fixed root-layering $(R_1, R_2, \ldots, R_\ell)$ can be written as

$$\prod_{k=1}^{\ell} \prod_{i \in R_k} \hat{w}_i \big( R_{k-1}, R_{1:(k-1)} \big). \quad (2)$$

The sampling algorithm starts by drawing a root-layering with a probability proportional to the total score of DAGs on them. Then, the parents of the nodes are sampled conditionally to the given the root-layering. The latter step is straightforward in their algorithm, since the parent set choices are independent of each other given the root-layering [Kuipers and Moffa, 2015, 2017].

The more involved part is sampling the root-layering. Suppose that we know the first $k$ layers of the root-layering. Then, the probability of next layer being $R_{k+1}$ is proportional to

$$f(R_{k+1}, N \setminus R_{1:k}) \prod_{i \in R_{k+1}} \hat{w}_i(R_k, R_{1:k}), \quad (3)$$

where

$$f(R_{k+1}, U) := \sum_{\substack{R_{k+2}, \ldots, R_\ell \\ R_{k+1:\ell} = U \\ R_j \text{ are disjoint}}} \prod_{j=k+2}^{\ell} \prod_{i \in R_j} \hat{w}_i(R_{j-1}, N \setminus R_{j:\ell}).$$

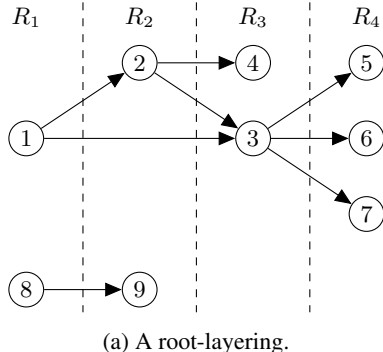

(a) A root-layering.

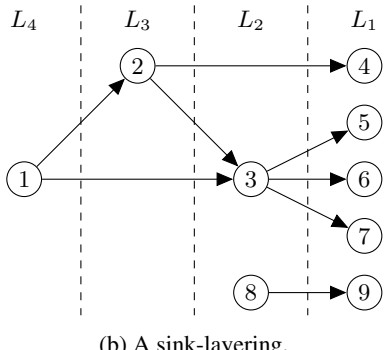

(b) A sink-layering.

Figure 1: The root-layering and the sink-layering of the same DAG.

The value of $f$ equals the total score of parent set choices for the remaining nodes $U \setminus R_{k+1}$ if $R_{k+1}$ is chosen to be the $(k+1)$th layer. In other words, the formula considers all possible extensions for the partial root-layering $(R_1, R_2, \ldots, R_{k+1})$ and sums up their scores.

By an inclusion–exclusion argument, Talvitie et al. [2019] observe that $f(R, U)$ equals

$$\sum_{S \subseteq (U \setminus R)} (-1)^{|(U \setminus R) \setminus S|} \left( \prod_{i \in (U \setminus R) \setminus S} \hat{w}_i(N \setminus U) \right) g(S),$$

where $g$ is defined recursively by

$$g(U) = \sum_{\emptyset \neq R \subseteq U} (-1)^{|R|+1} g(U \setminus R) \prod_{i \in R} \hat{w}_i(N \setminus U)$$

and $g(\emptyset) = 1$, inspired by Equation (1).

After precomputing the values of $g$ in time $O(3^n n)$, we can obtain all values of $f(R, U)$ for a fixed $U$ in time $O(2^{|U|}|U|^2)$ by using fast subset convolution [Björklund et al., 2007]. Faster $O(2^{|U|}|U|)$-time computation [Yates, 1937, Kennes and Smets, 1990] is achieved by observing that $f$ can be written as a product of the vector $[g(S)]_{S \subseteq U}$ and a Kronecker product of $|U|$ matrices of size $2 \times 2$, as noted by Talvitie et al. [2019].

Algorithm 1 describes the sampling procedure. Until every node belongs to some layer, a new layer $R_{k+1}$ is sampled with probabilities proportional to the weights described by Equation (3). Then, we sample the parents of the nodes in $R_{k+1}$ such that they belong to the set $R_{1:k}$ with at least one parent coming from the previous layer $R_k$. This results in a running time $O(2^n n)$ per sample.

It is possible to precompute all $3^n$ values of $f$ in time $O(3^n n)$, but this would lead to a poor space complexity: for $n = 20$, we would need to store at least 3 billion values to the memory. Thus, we argue that it is better to compute the values on the fly when they are needed, as this does not worsen the asymptotical time complexity.

---

**Algorithm 1:** Perfect sampling with root-layerings

$U \leftarrow N, k \leftarrow 0, R_0 \leftarrow \emptyset$;
**while** $U \neq \emptyset$ **do**
    Compute $f(R, U)$ for all $R \subseteq U$;
    weight$(R) \leftarrow f(R, U)$ for all $R \subseteq U$;
    weight$(\emptyset) \leftarrow 0$;
    **for** $\emptyset \neq R \subseteq U$ **do**
        **for** $i \in R$ **do**
            weight$(R) \leftarrow$ weight$(R) \cdot \hat{w}_i(R_k, R_{1:k})$;
    Draw $R_{k+1}$ proportionally to weight$(R_{k+1})$;
    **for** $i \in R_{k+1}$ **do**
        Draw $D_i \subseteq R_{1:k}$ with $D_i \cap R_k \neq \emptyset$
        proportionally to $w_i(D_i)$;
    $U \leftarrow U \setminus R_{k+1}$;
    $k \leftarrow k + 1$;
**return** the DAG $D$;

---

## 3 FASTER SAMPLING

Splitting a set of objects—like nodes or edges—in two halves and then performing computations over these smaller sets is a common algorithm design paradigm. Inspired by this, we seek to speed up sampling by partitioning the node set into two smaller sampling problems. For one of these problems, we will use Algorithm 1 of Talvitie et al. [2019], but for the other one we need a new algorithm that samples layers of sinks instead of source nodes. Roughly speaking, the reason for this is that one of the halves is not allowed to have parents from the other half, but the values of $f(R, U)$ are impacted by all nodes of the graph. We start the section by developing the sink-based algorithm, and then combine the two algorithms into an asymptotically faster one.

### 3.1 SAMPLING SINKS

Instead of dealing with root-layerings, we utilize *sink-layerings* proposed by Harviainen and Koivisto [2023] for a parameterized version of the problem. However, applying

their algorithm directly would require $O\big(4^n \operatorname{poly}(n)\big)$ time and space, so we need to optimize their method.

In a sink-layering $L_1, L_2, \ldots, L_\ell$ of a DAG $D$, the first layer $L_1$ contains the sinks of $D$, $L_2$ the sinks of $D[N \setminus L_1]$, and so forth. Thus, the layers are characterized by the length of the longest path to a node in $L_1$. This is illustrated in Figure 1b.

Similarly to root-layerings, our goal is to construct the sample by first drawing the layer $L_1$, then $L_2$, and so on. For sampling the layers, we need to know the total score of DAGs on $V \subseteq N$ whose set of sinks is $L$, denoted by $r(L, V)$. This quantity is hard to compute directly, so we instead compute its relaxed version $\check{r}(L, V)$ where we require $L$ to be only a subset of the sinks, obtained as

$$\check{r}(L, V) = h(V \setminus L) \prod_{i \in L} \hat{w}_i(V \setminus L).$$

Then, we find $r(L, V)$ by applying the Möbius inversion formula over supersets of $L$ in time $O(2^{|V|}|V|)$.

The layer $L_1$ can be sampled by just using the values $r(L, V)$, but sampling consequent layers is more complicated. In addition to requiring $L_{k+1}$ to be the set of sinks of $D[L_{k+1:\ell}]$, every node in $L_{k+1}$ must have a child in $L_k$, since otherwise it would be a sink of $D[L_{k:\ell}]$. Therefore, the parent sets of the nodes in $L_k$ must *cover* the nodes $L_{k+1}$.

When sampling $L_{k+1}$, we thus need to multiply $r(L_{k+1}, V)$ by the total score of parent set choices for the nodes in $L_k$ that cover $L_{k+1}$. We denote this quantity by $c(L_{k+1}, L_k, V)$, and it equals

$$\sum_{(D_i \subseteq V)_{i \in L_k}} \left[\!\!\left[ \bigcup_{i \in L_k} D_i \supseteq L_{k+1} \right]\!\!\right] \prod_{i \in L_k} w_i(D_i),$$

where $[\![X]\!]$ evaluates to 1 if and only if $X$ is true. This can be rewritten as

$$\sum_{S \supseteq L_{k+1}} \sum_{(D_i \subseteq V)_{i \in L_k}} \left[\!\!\left[ \bigcup_{i \in L_k} D_i = S \right]\!\!\right] \prod_{i \in L_k} w_i(D_i).$$

Now, the inner sum is a covering product over $|L_k|$ functions and can be computed for all $S \subseteq V$ with $O(2^{|V|}|V|)$ operations [Björklund et al., 2007]. The outer sum is a zeta transform over these values. Hence, we can sample $L_{k+1}$ with probabilities proportional to

$$r(L_{k+1}, V) \cdot c(L_{k+1}, L_k, V)$$

in time $O(2^{|V|}|V|)$.

Sampling the parents for the nodes in $L_k$ is made harder by that the parent sets must cover the nodes of $L_{k+1}$. We solve this by observing that the probability that the parent set of $i \in L_k$ is $D_i \subseteq V$ is proportional to

$$w_i(D_i) \cdot c(L_{k+1} \setminus D_i, L_k \setminus \{i\}, V),$$

after which the problem reduces to sampling the parents of $L_k \setminus \{i\}$ that cover $L_{k+1} \setminus D_i$. Thus, sampling the parents of all nodes in $L_k$ requires computing the values of $c$ with $|L_{k+1}|$ different arguments.

Algorithm 2 gives a high-level description of the implementation. By observing that

$$L_{1:1} \subsetneq L_{1:2} \subsetneq \cdots \subsetneq L_{1:\ell},$$

we obtain a time complexity $O(2^n n)$ per sample if the normalizing constants $h(U)$ have been precomputed. By combining Algorithm 2 with fast computation of the values of $h$, we get the following corollary:

**Corollary 1.** *Suppose there is an algorithm that computes all values of $h(U)$ in time $O(t(n))$. Then, DAG SAMPLING can be solved in preprocessing time $O(t(n))$ and sampling time $O(2^n n)$.*

This is the first time that the above speedup has been noted to our knowledge, since algorithms from previous work have been unable to utilize the precomputed normalizing constants.

---

**Algorithm 2:** Perfect sampling with sink-layerings

$V \leftarrow N, k \leftarrow 0, L_0 \leftarrow \emptyset$;
**while** $V \neq \emptyset$ **do**
    Compute $r(L, V)$ for all $L \subseteq V$;
    $\text{weight}(L) \leftarrow r(L, V)$ for all $L \subseteq V$;
    $\text{weight}(\emptyset) \leftarrow 0$;
    **if** $k > 0$ **then**
        Compute $c(L, L_k, V)$ for all $L \subseteq V$;
        **for** $L \subseteq V$ **do**
            $\text{weight}(L) \leftarrow \text{weight}(L) \cdot c(L, L_k, V)$;
    Draw $L_{k+1}$ proportionally to $\text{weight}(L)$;
    $L'_k \leftarrow L_k, L'_{k+1} \leftarrow L_{k+1}$;
    **for** $i \in L_k$ **do**
        $L'_k \leftarrow L'_k \setminus \{i\}$;
        Compute $c(L'_{k+1} \setminus D_i, L'_k, V)$ for all $D_i \subseteq V$;
        Draw $D_i$ proportionally to
            $w_i(D_i) \cdot c(L'_{k+1} \setminus D_i, L'_k, V)$;
        $L'_{k+1} \leftarrow L'_{k+1} \setminus D_i$;
    $V \leftarrow V \setminus L_{k+1}$;
    $k \leftarrow k + 1$;
**return** the DAG $D$;

---

### 3.1.1 Application to Ancestral Constraints

Perhaps surprisingly, Algorithm 2 enables perfect sampling of DAGs with a directed path from a given node $i$ to a given node $j$, which is not a modular feature. Such a constraint modelling (in)direct causation can, for example, be provided by an expert to allow ignoring network structures that are

clearly incorrect. Chen et al. [2015] and later Pensar et al. [2020] have given algorithms for computing the total score of DAGs where a path exists between the given nodes. However, a sampling algorithm for such DAGs has not been suggested before, possibly because of the lack of earlier sink-based sampling algorithms. We refer to this sampling problem as DAG SAMPLING WITH PATH.

Pensar et al. [2020] observe that partitioning the nodes into descendants and non-descendants of $i$ provides a method for computing the normalizing constant. Denote the set of descendants of $i$ by $U$ with $j \in U$. Then, the total score of DAGs with that descendant set of $i$ is

$$h\big(N \setminus (U \cup \{i\})\big) \cdot \hat{w}_i\big(N \setminus (U \cup \{i\})\big) \cdot f\big(\{i\}, U \cup \{i\}\big),$$

because both $N \setminus (U \cup \{i\})$ and $U \cup \{i\}$ must induce a DAG such that $i$ is the only source node of the latter. Additionally, the nodes in $U \cup \{i\}$ can have parents from $N \setminus (U \cup \{i\})$.

Thus, we can sample a DAG with a path from $i$ to $j$ by first sampling $U$ proportionally to the above formula, and then sample DAGs from $N \setminus (U \cup \{i\})$ and $U \cup \{i\}$. More precisely, we call Algorithm 2 on $N \setminus (U \cup \{i\})$, and Algorithm 1 on $U$ after initializing $R_0 = N \setminus (U \cup \{i\})$ and $R_1 = \{i\}$. Consequently, we get the following result:

**Theorem 2.** DAG SAMPLING WITH PATH *can be solved in preprocessing time* $O(3^n n)$ *and sampling time* $O(2^n n)$.

## 3.2 SPLIT IN TWO HALVES

We are now ready to combine algorithms 1 and 2 into a single algorithm. Our approach is based on the simple observation that for every DAG $D$, there is at least one subset $U \subseteq N$ of size $n/2$ that matches the $n/2$ last nodes in some topological order of $D$. We assume $n$ to be even for notational convenience, but the results extend straightforwardly for odd $n$. For a set $U$, denote the set of all DAGs with such a topological order by $\mathcal{D}(U)$, and observe that the total score of all DAGs in $\mathcal{D}(U)$ can be written as

$$q(U) := h(N \setminus U) \sum_{\emptyset \neq R_1 \subseteq U} f(R_1, U) \prod_{i \in R_1} \hat{w}_i(N \setminus U) :$$

for any DAG in $\mathcal{D}(U)$, the value $h(N \setminus U)$ includes the local scores of the nodes $N \setminus U$ as a term, $f(R_1, U)$ the local scores of $U \setminus R_1$, and $\prod_{i \in R_1} \hat{w}_i(N \setminus U)$ the local scores of $R_1$.

Suppose we have computed the total score of all DAGs in $\mathcal{D}(U)$ for each $U$. We then sample a DAG by first picking the set $U$ proportionally to those scores, and afterwards draw a DAG $D$ from $\mathcal{D}(U)$ proportionally to $w(D)$. When sampling $D$ from $\mathcal{D}(U)$, it suffices to sample a DAG on the $n/2$ nodes $N \setminus U$ as well as a DAG on the $n/2$ nodes $U$ with the addition that the nodes in $U$ may have parents from $N \setminus U$. For the nodes in $U$, we run Algorithm 1, but give

$U$ as an argument and initialize $R_0$ to $N \setminus U$. Similarly, we sample a DAG on $N \setminus U$ by utilizing Algorithm 2.

Unfortunately, the above method does not yet sample DAGs proportionally to $w(D)$, since there can be multiple sets $U$ for which $D \in \mathcal{D}(U)$, making sampling such DAGs more likely. We solve this issue with rejection sampling by developing an algorithm that associates $D$ with exactly one set $U$ for which $D \in \mathcal{D}(U)$. After sampling the set $U$ and the DAG $D$, we *accept* $D$ if $U$ is the subset of nodes of size $n/2$ associated with $D$, and otherwise we *reject* the sample. Then, the distribution of accepted DAGs $D$ is proportional to $w(D)$ as desired. One possible test for accepting the DAG is described in Algorithm 3, but any deterministic mapping suffices.

---
**Algorithm 3:** Acceptance test

---
children$(i) \leftarrow \{j : i \in D_j\}$ for all $i \in N$;
stack $\leftarrow []$;
**for** $i \in N$ **do**
  **if** children$(i) = \emptyset$ **then**
    stack.push$(i)$;
**for** $|U|$ times **do**
  $i \leftarrow$ stack.pop$()$;
  **if** $i \notin U$ **then**
    **return** `reject`;
  **for** $j \in D_i$ **do**
    children$(j) \leftarrow$ children$(j) \setminus \{i\}$;
    **if** children$(j) = \emptyset$ **then**
      stack.push$(j)$;
**return** `accept`;

---

The three presented algorithms are combined into a single sampler in Algorithm 4. Because we need to brute-force the values of $h(U)$ and $g(U)$ only for sets $U$ of at most $n/2$ nodes, we get that the time complexity of precomputation is

$$\sum_{\substack{U \subseteq N \\ |U| \leq n/2}} O\big(2^{|U|}|U|\big) = O\big(2^{3n/2} \cdot \sqrt{n}\big).$$

Since both algorithms 1 and 2 are called on $n/2$ nodes, the running time of one call of the algorithm after the precomputation is seemingly $O(2^{n/2} n)$. However, we still need to optimize the sampling of the parent sets in Algorithm 1 to achieve that complexity, since otherwise it may take time $O(2^n n)$ at worst. In other words, we need to draw each parent set in time $O(2^{n/2} n)$ out of $O(2^n)$ possibilities. We achieve this with the inclusion–exclusion principle.

First, order the nodes in $R_k$ arbitrarily and partition the family of potential parent sets $D_i$ based on the smallest node from $R_k$ included in $D_i$. After picking the set of potential parent sets whose smallest node from $R_k$ is $j$, it remains to draw the rest of $D_i$ from

$$R^* := R_{0:k-1} \cup \{v \in R_k : v > j\}.$$

Suppose we have decided that the nodes $A \subseteq R^* \cup \{j\}$ with $j \in A$ should be included in $D_i$ and that $B \subseteq R^*$ should not be. Then, the total score of the parent sets that include some node $v \in R^* \setminus (A \cup B)$ is

$$\sum_{S \subseteq A \cup \{v\}} (-1)^{|S|} \cdot \hat{w}_i\big((R^* \cup \{v\}) \setminus (B \cup S)\big)$$

and the total score of those not including $v$ is

$$\sum_{S \subseteq A} (-1)^{|S|} \cdot \hat{w}_i\big(R^* \setminus (B \cup S)\big).$$

These give the unnormalized probability masses for choosing whether to include $v$ into $A$ or $B$. After $R^* \setminus (A \cup B)$ is of size at most $|R^*|/2$, we can iterate over all potential parent sets $D_i \subseteq R^* \cup \{j\}$ with $A \subseteq D_i$ and $D_i \cap B = \emptyset$ in $O(2^{n/2}n)$ time. Similarly, the inclusion–exclusion formulas take $O(2^{n/2})$ time to evaluate as long as $|A| \leq n/2$, which clearly holds.

---

**Algorithm 4:** Fast Sampling

---

**if** the algorithm is called for the first time **then**
  Compute $\hat{w}_i(U)$ for all $U \subseteq N$ and $i \in N$;
  Compute $h(U)$ for all $U \subseteq N$ with $|U| \leq n/2$;
  Compute $g(U)$ for all $U \subseteq N$ with $|U| \leq n/2$;
  Compute $q(U)$ for all $U \subseteq N$ with $|U| = n/2$;
Draw $U$ proportionally to $q(U)$;
Call Algorithm 1 with $R_0 = N \setminus U$;
Call Algorithm 2 with $V = N \setminus U$;
Call Algorithm 3 on the sampled DAG $D$ and $U$;
**if** $D$ is rejected **then**
  Restart the algorithm;
**return** $D$;

---

Finally, we need to consider the impact of having to restart the algorithm because of some DAGs appearing in multiple sets $\mathcal{D}(U)$. At worst, a DAG can be in the set $\mathcal{D}(U)$ for each $U$, which happens with an empty graph. Consequently, the worst-case expected time requirement for sampling is $O(2^{3n/2}n) = O(2.829^n)$, which occurs if the only positive local scores are for empty parent sets. More formally,

**Theorem 3.** DAG SAMPLING *can be solved in expected running time $O(2.829^n)$.*

Fortunately, the number of duplicates is much smaller on average over all DAGs, which we will prove next. After that, we give empirical evidence that similar holds even when the parent sets are more constrained. Recall that $D$ encodes a partial order $P$. Let $ij \in P$ if there is a directed path from $i$ to $j$ in $D$. This relation is reflexive, antisymmetric, and transitive. An ideal $I \subseteq N$ of a DAG is a subset of nodes such that if $j \in I$ and $ij \in P$, then $i \in I$. In other words, the ancestors of a node in the ideal must also be in the ideal. If $D \in \mathcal{D}(U)$, then $N \setminus U$ is an ideal of $D$.

**Lemma 4.** *As $n$ tends to infinity, the DAGs of $n$ nodes have fewer than $1.742$ ideals of size $n/2$ on average.*

*Proof.* Let $G(n)$ be the number of DAGs of $n$ nodes. These values obey an asymptotical formula

$$G(n) \sim C 2^{\binom{n}{2}} n! (-\alpha)^{-n}$$

with $\alpha \approx -1.488$ and $C \approx 1.741$ given by Stanley [1973].

Observe that there are $2^{n^2/4}$ possible subsets of edges from $N \setminus U$ to $U$. Thus, the average number of ideals is

$$G(|N|)^{-1} \sum_{\substack{U \subseteq N \\ |U| = n/2}} 2^{n^2/4} \cdot G(|U|) \cdot G(|N \setminus U|).$$

As $n$ increases, $G(n)^{-1}G(n/2)^2$ approaches

$$C \cdot \binom{n}{n/2}^{-1} 2^{-n^2/4},$$

and so the average tends to $C < 1.742$. $\square$

Insisting that every node in $U$ has a parent slightly reduces the number of restarts. DAGs with fewer than $n/2$ nodes with parents are handled as a special case: if $U$ is the set of nodes with parents, then the total score of such DAGs is

$$\left(\prod_{i \in N \setminus U} w_i(\emptyset)\right) \sum_{\emptyset \neq R_1 \subseteq U} f(R_1, U) \prod_{i \in R_1} \hat{w}_i(N \setminus U).$$

It should be noted that most DAGs are dense, so they will dominate the sum in computing the average over all DAGs. On the other hand, most score functions penalize large parent sets. Still, the average-case analysis gives us hope that the worst-case complexity might not be what occurs in practice. We proceed to verify this empirically.

# 4 EMPIRICAL RESULTS

We start by discussing implementation details. Then, we report the results from our experiments.

## 4.1 NUMERICAL STABILITY

Efficient implementation of the presented algorithms requires the use of subtraction, which may lead to issues with numerical stability like catastrophic cancellation. The potential issue with stability was observed by Talvitie et al. [2019], and although they were able to make the computation monotone, the preprocessing time and the space complexity increased significantly as a consequence.

On the other hand, if the numerical operations are performed over integers of sufficiently many bits, we can avoid

Table 2: Preprocessing times of the algorithms and the average times for sampling one network structure, reported in seconds.

(a) Uniform sampling with randomized potential parent sets.

| | Talvitie et al. [2019] | | *this paper* | |
|---|---|---|---|---|
| $n$ | Preprocessing | Sampling | Preprocessing | Sampling |
| 15 | $1 \cdot 10^0$ | $1 \cdot 10^{-2}$ | $\mathbf{5 \cdot 10^{-1}}$ | $\mathbf{3 \cdot 10^{-4}}$ |
| 16 | $4 \cdot 10^0$ | $2 \cdot 10^{-2}$ | $\mathbf{2 \cdot 10^{0}}$ | $\mathbf{4 \cdot 10^{-4}}$ |
| 17 | $1 \cdot 10^1$ | $5 \cdot 10^{-2}$ | $\mathbf{4 \cdot 10^{0}}$ | $\mathbf{8 \cdot 10^{-4}}$ |
| 18 | $3 \cdot 10^1$ | $1 \cdot 10^{-1}$ | $\mathbf{1 \cdot 10^{1}}$ | $\mathbf{1 \cdot 10^{-3}}$ |
| 19 | $1 \cdot 10^2$ | $3 \cdot 10^{-1}$ | $\mathbf{3 \cdot 10^{1}}$ | $\mathbf{2 \cdot 10^{-3}}$ |
| 20 | $3 \cdot 10^2$ | $6 \cdot 10^{-1}$ | $\mathbf{1 \cdot 10^{2}}$ | $\mathbf{2 \cdot 10^{-3}}$ |
| 21 | $1 \cdot 10^3$ | $1 \cdot 10^{0}$ | $\mathbf{3 \cdot 10^{2}}$ | $\mathbf{5 \cdot 10^{-3}}$ |

(b) Weighted sampling with parent sets of size at most 2.

| | Talvitie et al. [2019] | | *this paper* | |
|---|---|---|---|---|
| $n$ | Preprocessing | Sampling | Preprocessing | Sampling |
| 15 | $1 \cdot 10^0$ | $1 \cdot 10^{-2}$ | $\mathbf{6 \cdot 10^{-1}}$ | $\mathbf{1 \cdot 10^{-3}}$ |
| 16 | $4 \cdot 10^0$ | $3 \cdot 10^{-2}$ | $\mathbf{2 \cdot 10^{0}}$ | $\mathbf{2 \cdot 10^{-3}}$ |
| 17 | $1 \cdot 10^1$ | $6 \cdot 10^{-2}$ | $\mathbf{5 \cdot 10^{0}}$ | $\mathbf{4 \cdot 10^{-3}}$ |
| 18 | $4 \cdot 10^1$ | $1 \cdot 10^{-1}$ | $\mathbf{2 \cdot 10^{1}}$ | $\mathbf{5 \cdot 10^{-3}}$ |
| 19 | $1 \cdot 10^2$ | $3 \cdot 10^{-1}$ | $\mathbf{4 \cdot 10^{1}}$ | $\mathbf{1 \cdot 10^{-2}}$ |
| 20 | $3 \cdot 10^2$ | $6 \cdot 10^{-1}$ | $\mathbf{1 \cdot 10^{2}}$ | $\mathbf{2 \cdot 10^{-2}}$ |
| 21 | $1 \cdot 10^3$ | $1 \cdot 10^{0}$ | $\mathbf{4 \cdot 10^{2}}$ | $\mathbf{4 \cdot 10^{-2}}$ |

stability issues. Notice that each $\mathcal{D}(U)$ is of size at most $\left((n/2)!\right)^2 2^{n(n-1)/2}$ and there are roughly $2^n$ sets $U$. Letting $M = \max_i \max_{D_i} w_i(D_i)$ be the largest local score, we get that we need at most

$$\log_2\left(\left((n/2)!\right)^2 2^{n(n+1)/2} \cdot M^n\right) = O(n^2 + n\log M)$$

bits for representing any of the numbers.

If floating point numbers are preferred, there are several potential ways of mitigating stability issues, with the most obvious one being the use of numbers with more bits. Alternatively, one may look into rounding the local scores into integers to then perform the rest of the computations exactly. The best method likely varies depending on the use case and the local scores of the instance. However, a more detailed analysis of numerical stability and rounding techniques are out of the scope of the present work, as implementing them is an engineering task of its own.

### 4.2 IMPLEMENTATIONS

We compare our Algorithm 4 against the Algorithm 1 of Talvitie et al. [2019]. Since no publicly available implementation of the latter exists, we have implemented both algorithms in C++. Although the algorithms seem numerically stable when implemented with floating point numbers based on a small-scale experiment (Appendix A), we instead use 512-bit integers in the following experiments to ensure that both of them sample from the same posterior distribution. Because we restrict ourselves to integer-valued local scores for a fair comparison of the algorithms, we cannot evaluate them on the common benchmark instances.

The implementations used up to 10 GB of memory out of the available 16 GB. See supplementary materials for source codes and instructions on compiling the programs.

### 4.3 RESULTS

We next present the results of our experiments. In the first one, we consider sampling DAGs from a uniform distri-

bution with a randomized family of potential parent sets. More precisely, each local score is assigned either 0 or 1 uniformly at random. For each $n = 15, 16, \ldots, 21$, Table 2 reports the preprocessing time and the time for drawing one (accepted) sample as an average time over hundred samples. We see that our method draws samples up to two orders of magnitude faster than the algorithm of Talvitie et al. [2019] and achieves lower preprocessing time.

The average number of ideals of size $n/2$ increases as the maximum allowed number of parents of the nodes is decreased. In our second experiment, we bound the parent set size to 2, and pick random 8-bit scores for the parent sets. Like expected, sampling becomes slower for our algorithm as seen in Table 2b, but the rejection sampling method is still 10–30 times faster with the ratio increasing in $n$.

## 5 CONCLUDING REMARKS

We presented the first algorithm for sampling DAGs with the base of the exponent less than 3 in its time complexity. The result was achieved by considering a family of subsets of DAGs obtained by partitioning the set of nodes into two halves. The attentive reader may wonder if this is optimal: could a better complexity be achieved by partitioning the nodes unevenly or into more sets? Unfortunately, the answer seems to be negative. For partitions of varying size, the precomputation cost of either the values of $h$ or $g$ increases, leading to a worse complexity. On the other hand, partitioning the nodes into more than two sets often increases the amount of duplicate counting.

One potential method for improving the running time would be to discover a combinatorial upper bound for the total score of DAGs and then apply rejection sampling in a self-reducible manner—an approach that has worked in perfect sampling of weighted permutations [Huber, 2006]. Other open questions relate to mitigating the impact of non-monotone computation to numerical stability: does a monotone algorithm of similar complexity exist, or could for example rounding techniques be utilized without impacting the distribution of DAGs too much?

## Acknowledgements

Research partially supported by Research Council of Finland, Grant 351156.

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

# Faster Perfect Sampling of Bayesian Network Structures (Supplementary Material)

**Juha Harviainen**[1] **Mikko Koivisto**[1]

[1]Department of Computer Science, University of Helsinki, Helsinki, Finland

## A EXPERIMENT ON STABILITY

To get an intuition of the numerical stability, we ran a small experiment with a commonly used benchmark network of 11 nodes of Sachs et al. [2005]. The (non-pruned) BDeu local scores are computed from 1000 data points generated from the ground truth. We compare the posterior distributions of three instantiations of different algorithms: the implementation by Talvitie et al. [2019] of their monotone algorithm with sampling time $O(2^n n)$, our implementation of the non-monotone Algorithm 1 of Talvitie et al. [2019], and our Algorithm 4 based on rejection sampling.

Each instantiation was used to draw 10000 DAGs from the posterior distribution, whose scores were then plotted as a histogram in Figure 2. To check the consistency of the results, we repeated the experiment three times, illustrated by the shaded bars in the plot. The distributions obtained by both non-monotone instantiations are close to the monotone one, which suggests numerical stability with at least this benchmark instance. The stability on larger networks remains uncertain, since running the monotone algorithm quickly becomes infeasible as the number of nodes increases.

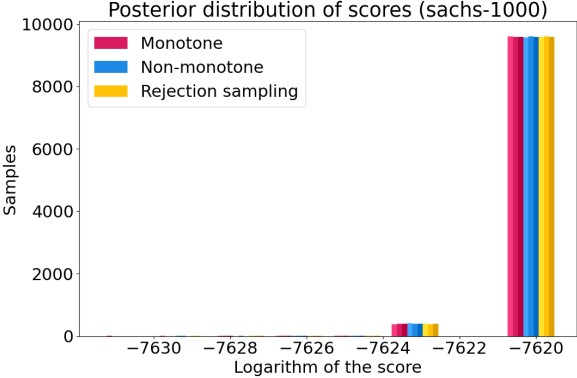

Figure 2: The distribution of the scores of DAGs sampled from the posterior distribution.