# OpenReview forum: "Faster Perfect Sampling of Bayesian Network Structures"
_auai.org/UAI/2024/Conference — UAI 2024 spotlight_

### Official Review · Reviewer_D9sp · 2024-03-20

**Q2-1 Originality-Novelty:** 3
**Q2-2 Correctness-Technical Quality:** 3
**Q2-5 Clarity Of Writing:** 4

**Q1 Summary And Contributions:**

The paper presents an algorithm for sampling bayesian network structures with a better asymptotic bound (O(2.829^n)). It provides a detailed description of existing work and build upon it to come up with a divide-and-rule kind algorithm for faster sampling of Bayesian network structures. It provides empirical evidence of speedup and provides a detailed account.

**Q2-3 Extent To Which Claims Are Supported By Evidence:**

3: Good: the main claims are supported by convincing evidence (in the form of adequate experimental evaluation, proofs, (pseudo-)code, references, assumptions).

**Q2-4 Reproducibility:**

3: Good: key resources (e.g. proofs, code, data) are available and key details (e.g. proofs, experimental setup) are sufficiently well-described for competent researchers to confidently reproduce the main results.

**Q3 Main Strengths:**

It is a well written paper. The theoretical advances have been described in good detail with intuition provided using figures. It seems to be a technically sound paper with a good description of related work. There are enough details provided for implementing their proposed algorithm.

**Q4 Main Weakness:**

1. It would really improve the reading of the paper if they demonstrate a dry run of their algorithm to sample a Bayesian Network structure with say 5 nodes.
2. Table 2 can be replaced with a graph that shows the trends as the number of nodes increase ('n') which would actually help to see if the claimed trend actually happens in practice or not. The numbers in the table do show that the proposed algorithm is better in practice but doesn't say anything about the trend.

**Q5 Detailed Comments To The Authors:**

1. A description of why fast-matrix multiplication is not good enough despite giving a better bound would be nice.
2. Experiments around sampling with an ancestral path constraint should add to the convincibility of the algorithm in that application.

**Q9 Complying With Reviewing Instructions:**

Yes

---

> ### Author Rebuttal · Authors · 2024-04-05
>
> Thank you for the positive review and the suggestions. We will consider methods for improving the presentation of the contributions and make it clearer that the issue with fast matrix multiplication is because of the large constant factors that are not visible in the asymptotic complexity.

---

### Official Review · Reviewer_C1pK · 2024-03-22

**Q2-1 Originality-Novelty:** 2
**Q2-2 Correctness-Technical Quality:** 3
**Q2-5 Clarity Of Writing:** 4

**Q1 Summary And Contributions:**

This work presents a novel method for perfect (or exact) sampling of BN structures from distributions proportional to decomposable scores.
The novel method is proven to improve the execution time, both theoretically and with experiments.
This task has implications for structure learning algorithms and causal discovery methods, as it allows for better Bayesian model averaging.

**Q2-3 Extent To Which Claims Are Supported By Evidence:**

3: Good: the main claims are supported by convincing evidence (in the form of adequate experimental evaluation, proofs, (pseudo-)code, references, assumptions).

**Q2-4 Reproducibility:**

3: Good: key resources (e.g. proofs, code, data) are available and key details (e.g. proofs, experimental setup) are sufficiently well-described for competent researchers to confidently reproduce the main results.

**Q3 Main Strengths:**

- clearly written even if technical
- code available to reproduce the experiments
- proofs of the results are in the main paper

**Q4 Main Weakness:**

- probably the subject of this work is relevant only to a small subset of the ML/UAI audience
- the contribution seems incremental over the prior literature

**Q5 Detailed Comments To The Authors:**

I think the paper could benefit from more applications of the developed method to causal discovery scenarios, this would make the work more appealing to a larger audience.

**Q9 Complying With Reviewing Instructions:**

Yes

---

> ### Author Rebuttal · Authors · 2024-04-05
>
> Thank you for the positive review.
>
> Although the improvement in the base of the exponent of preprocessing is not the largest, the sampling phase of our algorithm is significantly faster than in previous work. Since we are typically interested in drawing multiple DAGs rather than just one, we can draw several orders of magnitude more DAGs with the present method than with previous work, if the time budget is fixed. Additionally, we hope that our non-trivial rejection sampling construction can be improved in the future to attain even faster preprocessing time.
>
> We also think that UAI has certainly the right audience for the present work because of its long history in publishing results related to Baysian networks [Koivisto, 2006; Silander and Myllymäki, 2006; Tian and He, 2009], including recent work on sampling network structures [Talvitie et al., 2019; Harviainen and Koivisto, 2023].

---

### Official Review · Reviewer_Cb4S · 2024-03-24

**Q2-1 Originality-Novelty:** 3
**Q2-2 Correctness-Technical Quality:** 3
**Q2-5 Clarity Of Writing:** 4

**Q1 Summary And Contributions:**

The paper looks at the problem of producing a sample of Bayesian networks drawn from a distribution according to their posterior probably. The paper introduces a novel method for doing this which has lower time complexity than any existing known method. In addition to this method and associated proofs, the paper presents practical evaluation of the technique that shows it is faster than another recent method in practice as well as theory.

**Q2-3 Extent To Which Claims Are Supported By Evidence:**

4: Excellent: all claims are supported by very convincing evidence (in the form of comprehensive experimental evaluation, rigorous mathematical proofs, detailed (pseudo-)code, precise references, well-motivated and realistic assumptions) and the authors deliver what they promise.

**Q2-4 Reproducibility:**

4: Excellent: key resources (e.g. proofs, code, data) are available and key details (e.g. proof sketches, experimental setup) are comprehensively described for competent researchers to confidently and easily reproduce the main results.

**Q3 Main Strengths:**

The paper is highly relevant to the conference, being both about probabilistic models and probabilistic sampling of these models.

The paper makes a significant improvement on the state-of-the-art, both in theory and in practice. The time complexity is sub O(3^n) for the first time and the practical results show roughly a halving of pre-processing times and much larger decreases in sampling times.

The paper is very clearly written for something that is inherently very mathematical and precise. There is a clear introduction to all the required material; the presented algorithms help understanding of how the material in the text fits together; the mathematics is rooted in concepts familiar to the field, such as nodes and scores, rather than requiring transformations to abstract domains.

**Q4 Main Weakness:**

The experimentation is on some rather synthetic or non-standard datasets rather than well-known and well-studied ones. However, the rational for this is given and has merit.

The experimental results are adequate for the paper as the major contribution is the theoretical material, but they are not particularly extensive.

**Q5 Detailed Comments To The Authors:**

I assume the times in table 2 and the text are in seconds - this isn't explicitly stated anywhere. (Values given in Talvitie et al. would suggest they are seconds).

Given that the assessment of numerical stability reveals few practical problems with the technique, it seems that experiments with standard datasets could be presented in future work. At the very least, it would be useful to know whether the technique can actually be applied to these real(istic) datasets without numerical instability introducing bias.

**Q9 Complying With Reviewing Instructions:**

Yes

---

> ### Author Rebuttal · Authors · 2024-04-05
>
> Thank you for the positive review and the suggestions.
>
> You are correct that we had forgotten to include the unit of time in Table 2, and the running times are indeed reported in seconds. We will fix this to the next version of the paper.
>
> To give a better response to the concerns related to numerical stability, we have now performed an experiment on the Child network that consists of 20 variables. The number of variables is too large for methods that are guaranteed to be numerically stable, so we instead compared the posterior probabilities of edges computed by two distinct possibly unstable methods.
>
> More precisely, we generated 200 data points from the ground truth and computed local scores for parent sets of up to five nodes. Then, we used the (numerically unstable) code of Pensar et al. [2020] to compute the exact posterior probabilities of the edges, and sampled 10000 DAGs with our rejection sampling algorithm to estimate the same probabilities. The maximum of the absolute errors over all edge posterior probabilities was $0.0107$ and the average absolute error was $0.0005$, suggesting that the algorithms remain rather stable even with larger networks.

---

### Official Review · Reviewer_z9zC · 2024-03-25

**Q2-1 Originality-Novelty:** 2
**Q2-2 Correctness-Technical Quality:** 3
**Q2-5 Clarity Of Writing:** 2

**Q1 Summary And Contributions:**

This paper proposes a fast, practical algorithm to perform perfect sampling of Bayesian network structures. It improves the sampling efficiency by decomposing the sampling problem into two sub-problems and solving them accordingly.

**Q2-3 Extent To Which Claims Are Supported By Evidence:**

1: Poor: the authors fail to convincingly backup their main claims (e.g., if the experimental evaluation is flawed, proofs are lacking or invalid, references are missing, assumptions are not realistic, not specified, or not motivated).

**Q2-4 Reproducibility:**

1: Poor: key details (e.g. proof sketches, experimental setup) are incomplete/unclear, or key resources (e.g. proofs, code, data) are unavailable.

**Q3 Main Strengths:**

1. The paper aims to address long-existing, practical issues prevalent in the realm of Bayesian network structure learning.
2. The paper, based on the existing advances in performing Bayesian inference of DAG structure via dynamic programming, proposes to improve efficiency by splitting the sampling problems and employing sink-layering.

**Q4 Main Weakness:**

1. The presentation of this paper is confusing. It is hard to identify the contributions of this paper. I suggest the authors emphasize the contributions and their advantages.
2. One of my major concerns is that from Table 1 and all the theorems, I do not observe a significant improvement in complexity.
3. The authors mention the drawbacks are caused by the violation of certain assumptions (such as monotone) in empirical experiments. I suggest the author be explicit about the assumptions of the proposed algorithm.
4. Another major concern of mine is the limited empirical results, lack of comparisons, and wrongful use of the real Sachs dataset. I will be more specific in the detailed comments section.
5. All the theorems and corollary require detailed proof. If the theorems are introduced from other references, please provide citations. If the theoretical results are obvious, which in my opinion they are not, then it is unnecessary to formulate it as a theorem/corollary. I do not see any detailed proof for Theorem 1, 2, 3, and Corollary 1 in the main paper and supplementary.
6. I wonder if the authors submit the right supplementary version. They claim they provide the source codes and instructions for compiling the programs in section 4.2. However, I do not see these contents in the supplementary.

**Q5 Detailed Comments To The Authors:**

**Regarding Experiments:**
- The paper fails to provide any information on how the ground truth DAG and data are generated.
- Why the number of variables are set from 15 to 21? Is it because these are the experimental settings that the proposed algorithm outperforms the baseline? It feels like the authors are cherry-picking the experimental settings that the proposed algorithms perform well and draw conclusions based on partial results.
-  Is it reasonable to just compare to one baseline? I understand that algorithm 1 in Talvitie et al., 2019 is the main baseline that the paper wants to compare with. However, it is not comprehensive. I suggest the authors compare with other Bayesian inference methods of Bayesian network structure. Several Bayesian inference methods have emerged recently, such as MCMC sampling-based methods Kuipers et al., 2021, Viinikka et al., 2020, variational inference-based methods Lorch et al., 2021, Cundy et al., 2021, and so on.

> Kuipers, Jack, Polina Suter, and Giusi Moffa. "Efficient sampling and structure learning of Bayesian networks." Journal of Computational and Graphical Statistics 31.3 (2022): 639-650.

> Viinikka, Jussi, et al. "Towards scalable bayesian learning of causal dags." Advances in Neural Information Processing Systems 33 (2020): 6584-6594.

> Lorch, Lars, et al. "Dibs: Differentiable bayesian structure learning." Advances in Neural Information Processing Systems 34 (2021): 24111-24123.

> Cundy, Chris, Aditya Grover, and Stefano Ermon. "Bcd nets: Scalable variational approaches for bayesian causal discovery." Advances in Neural Information Processing Systems 34 (2021): 7095-7110.

- I suggest the authors not only compare the efficiency but also compare the accuracy of the sampled DAG. A method is not practical if the accuracy is greatly compromised in the expanse of efficiency.
- I suggest the authors not perform synthetic data analysis on real data Sachs. The Sachs is the benchmark real causal discovery dataset.

**Regarding Efficiency:**
- I am confused that in section 2.4, the paper states that the Talvitie et al., 2019 achieves a complexity of $\mathcal{O}(3^n n)$ on preprocessing time and $\mathcal{O}(2^n n)$ on sampling time. Then according to Theorem 2, the proposed method has the same complexity.
- According to Table 1, comparing the last row to the third row, it seems the proposed method can achieve better preprocessing complexity. However, the sampling complexity, best or worst, the proposed method achieves exponential complexity while Talvitie et al., 2019 achieves polynomial complexity. Hence it is hard to say that the proposed method is guaranteed to be better than Talvitie et al., 2019. It is possible for some values of $n$, the complexity of the proposed method is higher.

**Q9 Complying With Reviewing Instructions:**

Yes

---

> ### Author Rebuttal · Authors · 2024-04-05
>
> Thank you for the review.
>
> *“I do not observe a significant improvement in complexity*”
>
> We consider our work to have two important contributions. As a theoretical contribution, we get below the time complexity $O(3^n)$ for the first time and show that perfect sampling can be performed faster than what it takes to solve the #P-hard problem of exactly computing the marginals with the known algorithms. Empirically, our algorithm samples orders of magnitude more DAGs in the same time as the state of the art [Talvitie et al., 2019].
>
> *“All the theorems and corollary require detailed proof”*
>
> All theorems are proven in the paper. They formalize the results developed earlier in the text, and thus writing a separate proof after stating the theorems would lead to repetition. This approach improves the flow of the text and allows introducing ideas behind the algorithms while discussing their correctness and complexities.
>
> *”I wonder if the authors submit the right supplementary version”*
>
> Please note that the source codes are not in the appendix of the paper but as supplementary material on the OpenReview page.
>
> *”The paper fails to provide any information on how the ground truth DAG and data are generated”*
>
> For the synthetic instances, there is no ground truth DAG. Since our algorithm takes the number of nodes and the local scores as input, we directly generate the local scores as described in Section 4.3 to avoid numerical issues and ensure fair comparison of the implementations.
>
> For the local scores computed from a dataset, we provide an implementation with floating point numbers. The algorithm seems numerically rather stable based on initial experiments (Appendix A), but its stability on larger networks is harder to verify since the instances are intractable for numerically stable exact methods. See also our response to Reviewer Cb4S.
>
> *”Why the number of variables are set from 15 to 21”*
>
> Both algorithms solve instances with fewer than 15 variables in less than a second, meaning that the instances are rather easy. Since we improved the asymptotic complexity, it makes more sense to ask “how large instances can be solved in reasonable time”. Despite our focus on larger instances, our algorithm performs at least as well as the algorithm of Talvitie et al. on smaller instances, which the reviewer is welcome to verify with our implementations if they remain skeptical.
>
> *”Is it reasonable to just compare to one baseline”*
>
> The algorithm of Talvitie et al. is the only known exponential-time algorithm for perfect sampling of network structures until the present work. It is unclear how or why one even should compare a novel method to perfect sampling with methods for approximate sampling, as those algorithms solve a different problem. For example, if we want to compare the running times of our method and MCMC for drawing a single sample, then clearly one iteration of MCMC is faster than exact sampling, but for how many iterations should we run MCMC? Their mixing rates tend to be difficult to analyze, and so we have few guarantees that the samples would be drawn even roughly from the posterior distribution.
>
> *“I suggest the authors not only compare the efficiency but also compare the accuracy of the sampled DAG”*
>
> Since the motivation of sampling from the posterior is mitigating uncertainty with model averaging, it is unclear which metric the reviewer wants us to use for measuring the accuracy. For example, comparing the sampled structures to the ground truth (if it is even available) might not be reasonable if there is much uncertainty about the structure given the data.
>
> *“I suggest the authors not perform synthetic data analysis on real data Sachs”*
>
> Please note that we do not use Sachs as a benchmark for efficiency or accuracy (in terms of the ground truth), but rather as a preliminary verification that our algorithm is not too numerically unstable. It is the largest of [these](https://www.bnlearn.com/bnrepository/) benchmark networks for which the monotone algorithm of Talvitie et al. is feasible to run.
>
> *”according to Theorem 2, the proposed method has the same complexity [as that of Talvitie et al.]”*
>
> Theorem 2 discusses a variant of the problem to which the algorithm of Talvitie et al. is not applicable at all: the feature that a node $i$ is an ancestor of $j$ is not modular, so it cannot be enforced by manipulating the local scores.
>
> *“the proposed method achieves exponential [sampling] complexity while Talvitie et al., 2019 achieves polynomial complexity. Hence it is hard to say that the proposed method is guaranteed to be better”*
>
> It is true that after spending *significantly* more time and space to preprocessing, we can perfectly sample DAGs in polynomial time. However, the memory requirement quickly becomes infeasible, with 20 variables requiring storing $2 \cdot 10^{13}$ numbers to the memory. When the preprocessing and sampling time are considered together, our algorithm has the best known complexity.

---

### Official Review · Reviewer_h8wz · 2024-03-25

**Q2-1 Originality-Novelty:** 3
**Q2-2 Correctness-Technical Quality:** 3
**Q2-5 Clarity Of Writing:** 3

**Q1 Summary And Contributions:**

The paper presents a novel algorithm for sampling DAGs weighted by a given a modular score function. The first contribution is a faster sink layering sampling algorithm. This is further combined with a root layering sampling algorithm from Talvitie et al. and rejection sampling to obtain a perfect sampling approach with a complexity lower than O(3^n). Theoretical result for the expected rejection rate is given, and empirical runtime is also compared.

**Q2-3 Extent To Which Claims Are Supported By Evidence:**

3: Good: the main claims are supported by convincing evidence (in the form of adequate experimental evaluation, proofs, (pseudo-)code, references, assumptions).

**Q2-4 Reproducibility:**

4: Excellent: key resources (e.g. proofs, code, data) are available and key details (e.g. proof sketches, experimental setup) are comprehensively described for competent researchers to confidently and easily reproduce the main results.

**Q3 Main Strengths:**

1. The proposed approach improves on the state-of-the-art runtime for DAG sampling.
2. Theoretical result on the expected rejection sampling rate is given.
3. Empirical analysis shows order of magnitude improvement in runtime.

**Q4 Main Weakness:**

The theoretical results for the rejection sampling rate analyze the expected scenario but not the worst case.

**Q5 Detailed Comments To The Authors:**

Being not very familiar with the topic, I did find some of the mathematical notation a bit confusing. For example, $ w $ (with subscripts and hat) is used to represent the weight of a DAG, the local weight of a single node, as well as the zeta transform on a set of nodes.

**Q9 Complying With Reviewing Instructions:**

Yes

---

> ### Author Rebuttal · Authors · 2024-04-05
>
> Thank you for the positive review.
>
> We analyze the worst-case scenario right before Theorem 3: In instances where all probability mass is assigned on DAGs without edges, we need to draw $O(2^n)$ DAGs on average until getting an accepted DAG. The running time of the algorithm is a geometrically distributed random variable, so we cannot give it an upper bound that would always hold. However, the probability of repeatedly rejecting the samples reduces at an exponential rate in the number of samples, so the running time is unlikely to be notably larger than its expected value.

---

### Meta-Review · Area_Chair_B1W5 · 2024-04-16

This was a clearly written account of a clear advance on the state of the art. Only one reviewer was negative but I do not think their criticisms are valid (and seemed partly based on a misunderstanding of the nature of Bayesian inference).